# Selective Sorption of Heavy Metals by Renewable Polysaccharides

**DOI:** 10.3390/polym15224457

**Published:** 2023-11-18

**Authors:** Oshrat Levy-Ontman, Chanan Yanay, Yaron Alfi, Ofra Paz-Tal, Adi Wolfson

**Affiliations:** 1Department of Chemical and Green Engineering, Shamoon College of Engineering, Beer-Sheva 8434231, Israel; chanannn@gmail.com (C.Y.); adiw@sce.ac.il (A.W.); 2Nuclear Research Center, Negev, Beer-Sheva 8419001, Israel; alfiy79@gmail.com (Y.A.); ofrapt@gmail.com (O.P.-T.)

**Keywords:** adsorption, lanthanides, metals, polysaccharides, uranyl

## Abstract

Renewable and biodegradable polysaccharides have attracted interest for their wide applicability, among them their use as sorbents for heavy metal ions. Their high sorption capacity is due mainly to the acidic groups that populate the polysaccharide backbone, for example, carboxylic groups in alginate and sulfate ester groups in the iota and lambda carrageenans. In this study, these three polysaccharides were employed, alone or in different mixtures, to recover different heavy metal ions from aqueous solutions. All three polysaccharides were capable of adsorbing Eu^3+^, Sm^3+^, Er^3+^, or UO_2_^2+^ and their mixtures, findings that were also confirmed using XPS, TGA, and FTIR analyses. In addition, the highest sorption yields of all the metal ions were obtained using alginate, alone or in mixtures. While the alginate with carboxylic and hydroxyl groups adsorbed different ions with the same selectivity, carrageenans with sulfate ester and hydroxyl groups exhibited higher adsorption selectivity for lanthanides than for uranyl, indicating that the activity of the sulfate ester groups toward trivalent and smaller ions was higher.

## 1. Introduction

The recovery of heavy metals from wastewater generated during different industrial and institutional processes is of high importance [1]. This can be done using various techniques, including chemical precipitation [2], extraction [3], ion exchange [4], flotation [5], electrodialysis [6], membrane filtration [7], and sorption [8]. The latter was found to be one of the simplest and most effective methods for this purpose. Moreover, using adsorbents of biological origin, i.e., bioadsorbents, comprising either biomass or purified biomass-derived products such as polysaccharides is advantageous due to their biodegradability and renewability.

A variety of polysaccharides—long chains of carbohydrate molecules comprising different sugars adorned with different functional groups—have been used successfully as bioadsorbents to remove heavy metals from water solutions in recent years [9,10,11,12]. Besides their renewability, biodegradability, environmental friendliness, and wide availability, these biopolymers can easily adsorb a variety of metals of different sizes and valences due to the different acidic groups on their backbones, from sulfate ester groups to carboxylic and hydroxyl groups. Moreover, because the metal ion–polysaccharide composites can form hydrogels, the metals can also be easily removed from the solution and even recycled [13,14,15]. Polysaccharide-based hydrogels have also been prepared as beads by crosslinking polymer chains, either with metal ions or metal oxides, or by covalent bonding, and then employed as efficient adsorbents for a broad range of metal ions [16,17,18].

Recently, we also used a simple and straightforward method to separately remove europium ions (Eu^3+^) [19,20] and uranyl ions (UO_2_^2+^) [21] from water solutions by adding a polysaccharide solution to the aqueous metal solution, forming an insoluble hydrogel which, after mixing and adding ethanol, was removed. The studies were performed using several representative polysaccharides with different functional groups on their backbones: guar gum (*G*) with only hydroxyl groups; xanthan gum (*X*) with hydroxyl and carboxylic groups; and three carrageenans, lambda (λ), iota, (*C*), and kappa (κ), each with hydroxyl groups and 3, 2, and 1 sulfate ester units per subgroup, respectively. While all the tested polysaccharides adsorbed the metal ions, the more acidic carboxylic and sulfate ester groups augmented this adsorption. The sorption yield was also dictated by the polymer structure and the size and charge of the ions to be removed. In addition, we observed that the three carrageenans, which differed mainly in their sulfate ester contents, were superior Eu^3+^ sorbents [19,20], and that among the tested polysaccharides, *X* with carboxylic groups, exhibited the highest UO_2_^2+^ removal potential [21]. It was therefore suggested that not only could the different polysaccharides adsorb a variety of metal ions from aqueous mixtures, but also that the polysaccharide-based adsorbent system could be designed to selectively recover a specific metal ion from a solution of metal ions. Such selectivity was detected using alginic acid, chitosan, λ-carrageenan, and cellulose when the polysaccharides were immobilized on polyvinyl alcohol (PVA) and the mixture was dropped into a bead-forming solution of boric acid and glutaraldehyde [22]. In another study, different acidic polysaccharides, such as cross-linked alginic and pectic acids and their amide forms, were effective in the removal of Pb^2+^ and its separation from other metal ions [23].

As stated above, the unique acidic sulfate ester groups on the carrageenan backbone and their ability to form hydrogels can be very useful for adsorbing Eu^3+^ and UO_2_^2+^ from wastewater. Several reports have been published on the selective adsorption of heavy metals using renewable polysaccharide-based sorbents [22,24,25]. However, only limited information has been published on the metal selectivity of algal polysaccharides [25]. Moreover, to the best of our knowledge, the selective adsorption of lanthanides and actinides from multi-component mixtures by algal polysaccharides has not been investigated. Thus, in this study, we expanded on our previous studies and tested the sorption of other lanthanides, such as Sm^3+^ and Er^3+^, using carrageenans. In addition, as wastewaters usually contain several metal ions, the sorption of these ions from the solution and the selectivity of the sorption, which could be useful for the separation of the various metal ions, were also studied. We used three representative acidic polysaccharides with high acidic group contents—*I* and λ with sulfate ester groups and alginate (*A*) with carboxylic acid groups—for comparison, and studied the sorption of Eu^3+^, Sm^3+^, Er^3+^, and UO_2_^2+^, individually and in mixtures.

## 2. Materials and Methods

### 2.1. Materials

All the polysaccharides (λ—product #22049, batch #BCBP8978V; *I*—product # C1138, batch #SLCH6326; and *A*—product #2033, batch #098K1388), as well as europium chloride, samarium chloride, erbium chloride, uranyl acetate dihydrate, and other chemicals (analytical grade), were purchased from Sigma-Aldrich, Jerusalem, Israel.

### 2.2. Adsorbent Preparation and Utilization

The adsorbents were prepared and tested as follows:

#### 2.2.1. Preparation of the Polysaccharide Solutions

A total of 1 g of polysaccharide was dissolved in 100 mL of double-distilled water (DDW) and mixed for 1 h, yielding a 1 wt/v% aqueous solution of polysaccharide. When *I* or *A* was used, the mixture was heated to 50 °C and then stirred for an additional 1 h until a homogeneous solution was obtained.

#### 2.2.2. Sorption Experiments [19]

In a typical procedure, 3 mL of 1 wt/v% aqueous solution of polysaccharide (0.03 g) was added to 500, 700, or 1000 mg/L aqueous solutions of metal ions (Eu^3+^, Sm^3+^, Er^3+^, or UO_2_^2+^, alone or mixtures of two or three ions), to which was added water to yield a 10 mL solution that was mixed for 30 min at room temperature. Cold ethanol (4 × 10 mL) was then added to the aqueous solution. This was followed by mixing and removal of the solid via centrifugation and filtration. Finally, ethanol was evaporated from the solution and the concentration of the metal ion that was left in the solution was determined using ICP-OES (SPECTRO, model ARCOS, Spectro analytical instruments GmbH, Kleve, Germany). Each sorption experiment was performed five times, with a maximum standard deviation of ± 5%; the reported yields are the means of all the experiments. In addition, a control set with no polysaccharide and containing only metal ion solutions of appropriate concentrations at appropriate pH was run for comparison.

The sorption yield (Y) was calculated as follows:Y [%] = (C_in_ − C_f_)/C_in_ × 100
where C_in_ (mg/L) is the initial concentration of M^n+^ and C_f_ (mg/L) is the final concentration of M^n+^.

The specific metal selectivity (%) is defined as the ratio of the concentration of the specific metal that was adsorbed to the concentration of all metals adsorbed. For example, the specific metal selectivity for Eu^3+^ is:Selectivity (%) = (C_in_Eu^3+^ − C_f_ Eu^3+^)/((C_in_Eu^3+^ − C_f_ Eu^3+^) + (C_in_UO_2_^2+^ − C_f_ UO_2_^2+^))
where C_in_ (mg/L) is the initial concentration of M^n+^ and C_f_ (mg/L) is the final concentration of M^n+^.

### 2.3. Adsorbent Characterization after Sorption

The dried adsorbent that remained on the filter after sorption from a 1000 mg/L solution containing equal amounts of Eu^3+^, Er^3+^, and UO_2_^2+^ was dried for 24 h by lyophilization and then analyzed together with the pristine polysaccharides as detailed below.

### 2.4. Fourier-Transform Infrared (FTIR) Analysis

The pristine polysaccharides and the pellets obtained from the sorption experiments (after drying for 24 h under ambient conditions) were subjected to FTIR analysis using a Nicolet 6700 FTIR (Thermo Scientific, Waltham, MA, USA) spectrophotometer with an attenuated total reflectance (ATR) device outfitted with a diamond crystal plate. The recorded spectra were the means of 36 spectra taken in the wavenumber range of 650–4000 cm^−1^, with a 0.5 cm^−1^ resolution and atmospheric correction at room temperature (25 °C)

### 2.5. Thermal Gravimetric Analysis (TGA)

Post-sorption thermal changes in the polysaccharides were monitored using the TGA analyzer (TGA Q500 V20.13-Instrument, TA Instruments, New Castle, DE, USA) at a heating rate of 10 °C/min over a temperature range of 25–800 °C under a continuous nitrogen flow rate of 90 mL/min.

### 2.6. X-ray Photoelectron Spectroscopy (XPS)

XPS data were collected using an X-ray photoelectron spectrometer, the ESCALAB 250 ultrahigh vacuum (1 × 10^−9^ bar) apparatus with an AlKα X-ray source and a monochromator. The X-ray beam size was 500 μm and survey spectra were recorded with a pass energy (PE) of 150 eV, while high energy resolution spectra were recorded with a PE of 20 eV. To correct for charging effects, all spectra were calibrated relative to a carbon C 1s peak positioned at 284.8 eV. The XPS results were processed using the Advantage software, V6.6.

## 3. Results and Discussion

As noted above, because cation sorption to the different polysaccharides occurred preferentially via the acidic groups on the latter, we used polysaccharides with large numbers of acidic groups: *A*, a linear polymer that consists of L-glucuronic acid and D-mannuronic acid residues connected via 1,4-glycosidic linkages (Figure 1), with two carboxylic groups per polymer unit; *I*, a linear polymer that comprises D-galactose-4-sulfate and 3,6-anhydro-D-galactose-2-sulphate residues connected via 1,4-glycosidic linkages, with two sulfate ester groups per polymer unit (Figure 1); and λ, a linear polymer that contains chains of alternating 2-sulfated 1,3-linked α-d-galactose and 2,6-disulfated 1,4-linked β-d-galactose units, with three sulfate ester groups per polymer unit (Figure 1).

The investigation began by adding a solution of *A* or *I* to a solution of UO_2_^2+^, Eu^3+^, or Sm^3+^, followed by sorption of the metal ions under similar conditions (Table 1). For all three metal ions, the sorption yields for *A* exceeded those for *I* and were higher than previously published yields for other polysaccharides: *G*, *X*, λ, *I*, and κ [19,20,21]. This finding may be attributed to the high number of carboxylic groups on the backbone of *A* compared with those of *G* and xanthan gum *X*, which are also relatively more available compared with those in the backbone of *X* due to the simple structure of the polysaccharide chains. The availability of the carboxylic groups also confers on *A* the capacity to crosslink and form hydrogels with a variety of cations and to participate in hydrogel formation. In addition, the sorption yields for Sm^3+^, which, to the best of our knowledge, has not been tested before using any polysaccharide, were relatively similar to those for Eu^3+^ for both polysaccharides, all the more so when considering a standard deviation of ± 5%. The similar sorption yields of the two lanthanides can be explained by their very close ionic radii (0.964 and 0.950 Å for Sm^3+^ and Eu^3+^, respectively) [28] and similar electron configurations.

The metal ion sorption tests also showed that the sorption yield of UO_2_^2+^ on *A* is higher than that of Sm^3+^ or Eu^3+^ (*p* < 0.5 unpaired two-tailed Student’s *t*-test, Table 1).

Although the three radionuclides are known to form complexes with nitrogen- and oxygen-containing functional groups [29], the different sorption yields may be attributed to their multidentate inner-sphere complexes. A divalent ion such as uranyl needs only two carboxylic groups to coordinate, while trivalent ions such as lanthanides require three. Finally, as expected, reducing the concentration of the metal ion from 700 mg/L to 500 mg/L while maintaining the amount of polysaccharide at 0.03 g increased the sorption of all the metal ions on all the polysaccharides, as the number of coordinating sites on the polymer backbone increased relative to the number of metal ions.

Based on the above results, the two polysaccharides were also used for the first time to recover metal ions from a solution that contained both Eu^3+^ and Sm^3+^ in varying ratios, wherein the overall metal ion concentration in the solution was maintained at 700 mg/L (Table 2). As expected, the results in Table 2 show that the sorption yields from the solutions with the two metals were relatively similar, though consistently higher with *A* than with *I*. Moreover, ICP-OES analyses confirmed that polysaccharide selectivity for the two metal ions was similar, meaning that there was no competition between them, probably as a result of the similar valence, ionic radius, and electron configuration of each cation.

Based on these initial findings, we decided to study the sorption yields of solutions that contain different ratios of Eu^3+^ and UO_2_^2+^ (and whose overall metal concentrations are maintained at 1000 mg/L) on *A* and *I* (Table 3). We also performed these selectivity tests using λ, as previous results indicated that while λ formed hydrogels with Eu^3+^, no hydrogel was formed when it was added to the UO_2_^2+^ solution [19,20,21]. Published results showed that λ and trivalent ions like Al^3+^ and Fe^3+^ most likely form a six-coordinate complex via the hydroxyl/sulfate ester groups on the polysaccharide [30]. Not every trivalent cation, however, yielded a hydrogel with λ, e.g., Cr^3+^, as its ability to bind trivalent metal ions is determined by both the activation enthalpy and entropy. In this context, the electron configurations of Eu^3+^ and Sm^3+^ resemble those of Al^3+^ and Fe^3+^ but are dissimilar to those of Cr^3+^. It therefore seems that, like Al^3+^ and Fe^3+^, both metal ions can use their outer empty orbitals to hybridize [31].

The results in Table 3 show that *A* had the highest overall sorption yield for all three solutions containing different Eu^3+^ and UO_2_^2+^ concentrations, while *I* and λ had relatively similar but lower sorption yields. However, as was assumed, while *A* recovered both ions with the same selectivity, the adsorption selectivities of both *I* and λ were higher for Eu^3+^ than for UO_2_^2+^, a trend that was reflected in both carrageenans. Furthermore, changing the ratio of Eu^3+^ to UO_2_^2+^ in the initial solution only altered the adsorption selectivity slightly. In addition, although a hydrogel was not obtained when λ was mixed with UO_2_^+2^ alone, it was obtained when λ was added to a mixture of Eu^3+^ and UO_2_^2+^. Similar trends were also observed when Eu^3+^ was replaced with Er^3+^, a trivalent lanthanide whose removal from aqueous solutions using a polysaccharide has not, to the best of our knowledge, been tested before (Table 4).

Polysaccharide (*A*, *I,* and λ) performances were also tested using equal amounts of the three ions—UO_2_^2+^, Er^3+^, and Eu^3+^—at an overall concentration of 1000 mg/L. The overall yields and ion selectivity of the sorption are summarized in Table 5. As expected, the overall yield obtained using *A* was much higher than that obtained using either λ or *I*, and while sorption using *A* exhibited similar selectivities for the three cations—with a slight preference for UO_2_^2+^—both λ and *I* favored the sorption of Er^3+^ and Eu^3+^, and their respective selectivities for each metal ion were also similar.

### 3.1. FTIR Analysis

To gain more knowledge about the interactions between Eu^3+^, Er^3+^, and UO_2_^2+^ and the different polysaccharides (*I,* λ, and *A)*, the abundant surface functional groups of the three polysaccharides were determined using ATR-FTIR spectroscopy before and after sorption of Eu^3+^, Er^3+^, and UO_2_^2+^. The post-sorption spectra of the polysaccharide showed a significant difference compared with the relevant pristine polysaccharides *I,* λ, and *A*. The FTIR spectra of *I* and λ carrageenan polysaccharides after sorption of mixtures of Eu^3+^, Er^3+^, and UO_2_^2+^, noted as *I*-Er-Eu-U and λ-Er-Eu-U, respectively, were compared to their pristine polysaccharides, as shown in Figure 2a and Figure 2b, respectively.

As expected, the pristine carrageenans presented the same characteristic bands, indicating their basic chemical bonding as follows: peaks at 3200–3400 cm^−1^ due to the assigned O–H stretching vibrations; O–SO_3_ stretching vibration in D-galactose-4-sulfate in *I* and λ at 919 and 923 cm^−1^ and 851 and 844 cm^−1^, respectively, and at 808 and 810 cm^−1^, respectively, for D-galactose-2-sulfate [32,33]; and C–O stretching ascribed to the glycosidic linkage (3,6-anhydrogalactose) at 1067 and 1069 cm^−1^, respectively. Upon sorption of Eu^3+^, Er^3+^, and UO_2_^2+^ by the I and λ polysaccharides, the broad bands of the OH stretching vibrations shifted to lower wavenumbers (red-shift), from 3396 cm^−1^ to 3340 cm^−1^ and from 3415 cm^−1^ to 3349 cm^−1^, respectively. In addition, the bands were less intense than those observed in both pristine polysaccharides. Also, the H–O–H deformation bands of *I* shifted from 1646 cm^−1^ to 1624 cm^−1^ as observed previously after the sorption of Eu^3+^ [34]. These shifts and the reduction in band intensities could be attributed to the interaction with the metal ions, which reduced the hydrogen bonds upon complexation. Indeed, we observed that the interaction of uranyl [21] and europium ions [19] with the oxygen ion in the hydroxyl groups in *I* could lead to lower availability of the OH groups, which could otherwise self-assemble the polymer chains. Also, we can assume that, as expected, part of the metal ions interact with the hydroxyl groups in the polymeric chains, and thus, the selectivity for both lanthanides is not complete.

In the 1100–1300 cm^−1^ range, the characteristic O=S=O stretching band, which appears in pristine *I* and *I*-Er-Eu-U, shifted from 1212 cm^−1^ to 1175 cm^−1^, while for pristine λ, the band at 1222 cm^−1^ split into two bands at 1250 cm^−1^ and 1185 cm^−1^ in λ-Er-Eu-U [35]. Additionally, the O-SO_3_ stretching vibration in D-galactose-4-sulfate, which has a broad band at 919 cm^−1^ in pristine *I*, split into two sharp bands at 909 cm^−1^ and a small peak at 902 cm^−1^ in *I*-Er-Eu-U. These findings, which were also previously observed in *I*-UO_2_^2+^ [21], suggest that the sulfate ester groups are involved in the complexation of all the metal ions. Moreover, the intensities of the bands in the spectra were much lower than those in the spectra of the pristine polysaccharide, again confirming the interaction between the polymers and the various metals.

Figure 3 shows the spectrum of pristine *A* and its characteristic bands [36]. After the adsorption of the metal ions, however, the spectrum changed as follows: The broad band of pristine *A* assigned to the stretching vibrations of hydroxyl groups shifted from 3275 cm^−1^ to 3342 cm^−1^, indicating the involvement of OH groups in the metal ion interactions. In addition, the bands at 1588 cm^−1^, 1414 cm^−1^, and 1285 cm^−1^, assigned to the stretching vibrations of symmetric and asymmetric bands of carboxylate anions (O-C-O) shifted to bands at 1579 cm^−1^ (broad), 1427 cm^−1^, and 1312 cm^−1^, respectively. Also, the bands assigned to C-H stretching for the mannuronic acid functional group at wavenumber 873 cm^−1^ and for uronic acid at wavenumber 932 cm^−1^ overlapped to form one broad peak at 929 cm^−1^ in the *A*-Er-Eu-U spectrum. We assume that the broad bands that appeared in *A*-Eu-U-Er at 1579 cm^−1^ and at 929 cm^−1^ are probably due to the interaction between the three metal ions and the carbonyl group (C=O), which led to the deformation of the polysaccharide skeleton. Indeed, the bending vibration of δU-OH at 929 cm^−1^ [37], which indicated the presence of uranyl ion, was observed after the sorption of the metal ion mixture.

### 3.2. Thermal Gravimetric Analysis

Thermal gravimetric analysis (TGA) was performed to evaluate the thermal degradation behavior of *A/I*/λ-Er-Eu-U compared with their pristine polysaccharide analogs. Table 6 presents the TGA and TGA–DTA (differential thermal analysis) results for the polysaccharides (*A*/*I*/λ) before and after sorption, the percentage of mass loss at each stage (%wt loss), the onset temperature, and the differential thermogravimetric (DTG).

TGA results for pristine *I*, λ, and *A* are consistent with those of previous reports on the thermal behavior of the polysaccharides [20,21]. As expected, the initial weight loss of about 7–15% in all the polysaccharides observed in the first stage at temperatures below 200 °C was due to the evaporation of moisture (Table 6). This stage, which is mainly due to the interactions between the water molecules and the hydroxyl groups on the surface of the polymeric backbone, confirms the hydrophilic nature of the various polysaccharides. The second stage, at temperatures above 200 °C, involved several steps during which carbohydrate backbone fragmentation and de-polymerization occurred until their residues reached a constant weight loss (~70–80%) at up to ~350 °C for *A* and ~550 °C for *I* and λ. The higher residue value for the *A* carrageenan (~33%) compared with those of the *I* and λ carrageenans (~26%) indicates that the pristine *I* and λ carrageenans were less stable compared with *A* and that the degradation stage also involved the degradation of sulfate ester groups (-OSO_3_-) from the pendant chains attached to the polymeric backbone [38].

The post-sorption TGA and TGA–DTA thermograms of all the polysaccharides with the three ions differed from those of the pristine polysaccharides and the polysaccharides with one type of metal ion. A new stage in the 600–1000 °C temperature range was observed after sorption, which indicates that a chemical interaction occurred between the metal ions and the different functional groups in each polysaccharide. This stage was observed in the *I* and λ polysaccharides, whose residue values after sorption of the three ions were slightly higher than those of the pristine polysaccharides (from 26.30% to 27.49% and from 26.54% to 28.51%, respectively). This observation also confirmed the FTIR spectra. In contrast, the residue value for *A*-Er-Eu-U decreased from 33.45% to 26.67%, indicating that the chemical interactions between the metal ions and the carboxylic and hydroxyl groups were weak or that the polysaccharide morphology had changed.

As shown in Table 6, all the polysaccharides decomposed more easily after sorption: In the second stage (degradation step), the onset temperatures for *I*- and λ-Er-Eu-U decreased from 137.24 °C to 126.86 °C and from 181.62 °C to 137.60 °C, respectively. It seems that the -OSO_3_- groups in the pendant chains attached to the polymeric backbone were more sensitive to thermal degradation when they were attached to each of the metal ions, and therefore decomposed more easily. Furthermore, the onset temperature for *A*-Er-Eu-U decreased slightly from 150.67 °C to 149.64 °C, possibly due to weak binding between the carboxylic and hydroxy groups and the metal ions. However, in our previous study [20] of polysaccharide sorption behavior with different metal ions, we noticed that the amount of residue remaining after sorption was elevated: for example, the weight loss for *I*-Eu and λ-Eu increased from 26.54% for pristine *I* to 34.49% and from 26.34% for pristine λ to 34.25%, respectively [20]. Similar results were obtained for *I*-UO_2_^2+^ compared with pristine *I* (37.05% and 26.54%, respectively) [21]. The evidence indicates that although there is interaction between the -OSO_3_- group in the carrageenan and the Eu^3+^ and UO_2_^2+^ metal ions, the decomposition of the hydrogen bonds between the chains affects complex morphology and stability. Indeed, this observation is in agreement with the changes in the OH stretching range in the FTIR results, and the broad bands correspond to the -OSO_3_- group due to the sorption of Er^3+^, Eu^3+^, and UO_2_^2+^.

### 3.3. XPS Analysis

To elucidate the mode of sorption, the XPS spectra of *I* before and after adsorption were measured. The XPS survey spectrum of *I*-Er-Eu-U (Figure 4a) shows that all the metals adsorbed to *I* and were spectrally assigned to U4f7, Er4p, and Eu3d5. For both preparations, the deconvolution of S2p indicated two peaks (S2p3 Scan A and Scan B) (Figure 4c). The observed shift in the S2p3 Scan A peak for *I*-Er-Eu-U to lower binding energy may be attributed to a strong interaction between sulfur and Eu/Er. The ratio of S2p3 Scan A to Scan B (Figure 4c) was also observed to diverge in the two preparations, as the S2p3 Scan A spectrum was less dominant (12.88%) than that observed for *I* (46.74%). Deconvolution of the O1s peaks before and after adsorption revealed three peaks that are due to the existence of C–OH, C–O–C, and C=O bands (Figure 4b). The peak S2p3 scan C (assigned to OH) shifted to a lower energy band, suggesting that uranyl binds to the oxygen in the hydroxyl groups. The ratio of O1s Scan A to O1s Scan A/Scan B speaks was also altered following adsorption. It seems that, generally, the binding of the metals via oxygen effectively screened the OH by reducing the dipole forces in the matrix.

## 4. Conclusions

Polysaccharides can easily adsorb heavy metal ions in aqueous solutions via the functional groups on their backbones. Moreover, during the adsorption process and due to the physical crosslinking of the polysaccharide chains with metal ions, the complex precipitates as a hydrogel that can be easily removed from the solution. Herein, three representative polysaccharides—alginate (*A*) with carboxylic and hydroxyl groups and iota (*I*) and lambda (λ) with sulfate ester and hydroxyl groups—were successfully used to adsorb Eu^3+^, Sm^3+^, Er^3+^, and UO_2_^2+^, alone or as different mixtures; *A* had the highest sorption yields. Sorption from a binary or ternary mixture using *A* alone showed no selectivity, as was the case for *I* from a binary mixture of two lanthanides (Eu^3+^ and Sm^3+^). However, employing both *I* and λ carrageenans to adsorb a mixture of binary and ternary metal ions that comprised one or two lanthanides together with uranyl (Eu^3+^ and UO_2_^2+^, Er^3+^ and UO_2_^2+^ or Eu^3+^, Er^3+^, and UO_2_^2+^) showed higher selectivity for the trivalent and smaller lanthanides, indicating that the lanthanides have higher affinities for the sulfate ester groups in carrageenans. XPS analysis confirmed the presence of the metal ions in the polysaccharide hydrogels, and FTIR analysis showed that they interacted via the hydroxyl groups in all three polysaccharides, the carboxylic groups in *A*, and the sulfate ester groups in *I* and λ. TGA thermographs showed that the chemical interactions between the metal ions and the carboxylic and hydroxyl groups on *A* were weaker than those between the metal ions and the sulfate ester groups on both carrageenans, an observation that can explain the carrageenan’s sorption selectivity. Finally, as wastewaters usually contain several metal ions, selective sorption of one of the metal ions could be useful for separating the various metal ions, thus enabling better recovery of a specific ion.

## Figures and Tables

**Figure 1 polymers-15-04457-f001:**
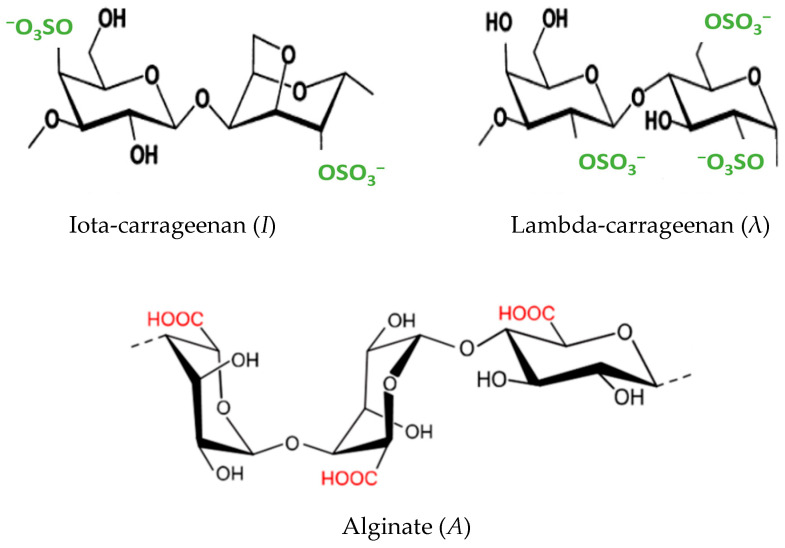
Structures of the three polysaccharides: Iota-carrageenan (*I*) [26], Lambda-carrageenan (λ) [26], and Alginate (*A*) [27].

**Figure 2 polymers-15-04457-f002:**
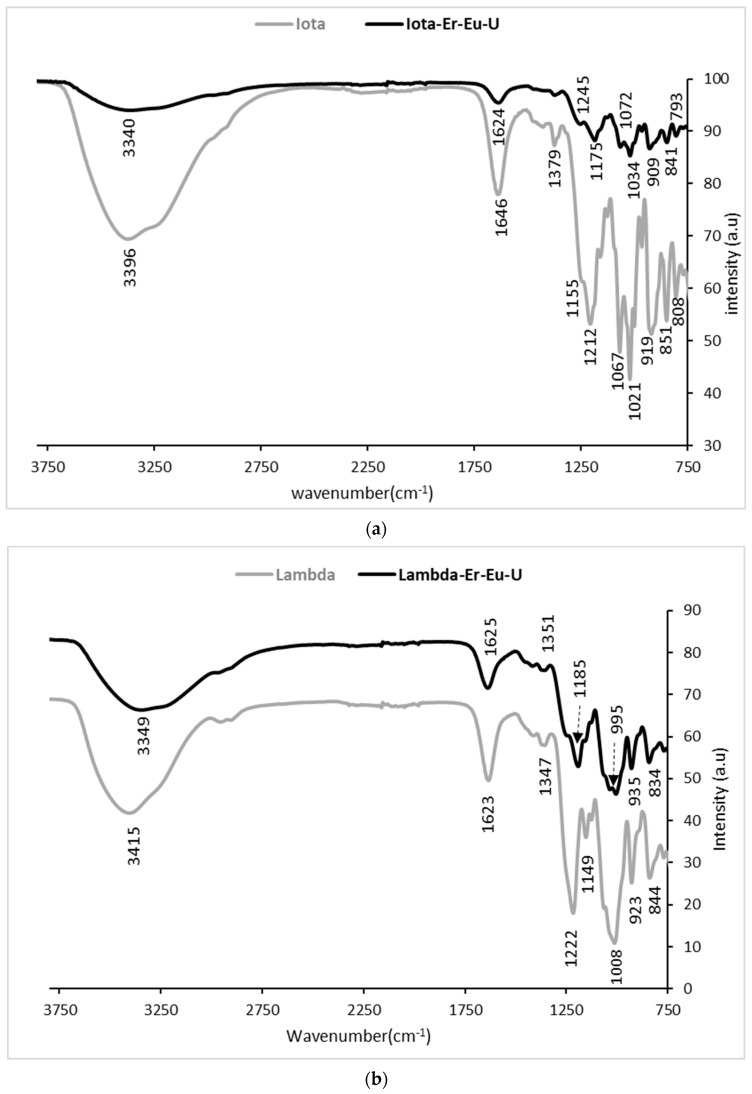
ATR-FTIR spectra of pristine *I* and *I*-Er-Eu-U (**a**), and pristine λ and λ-Er-Eu-U (**b**). Sorption conditions: 333 mg/L mixture of equal amounts of Er^3+^, Eu^3+^, and UO_2_^2+^; 3 mL of 1 wt/v% *I* or λ.

**Figure 3 polymers-15-04457-f003:**
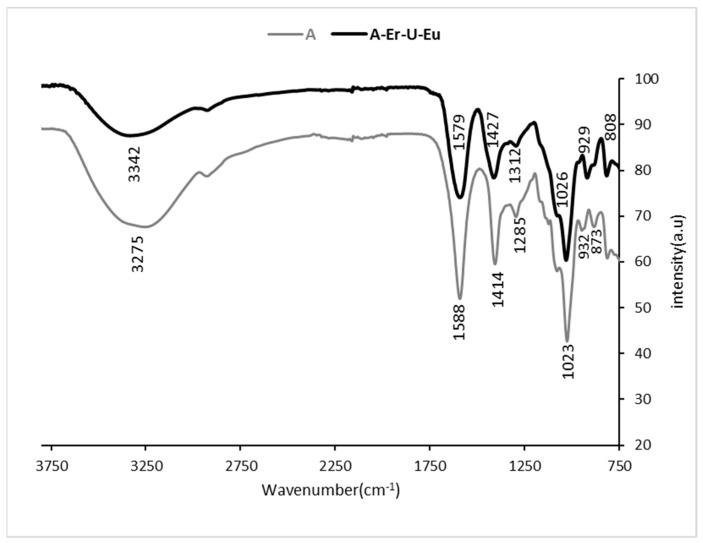
ATR-FTIR spectra of pristine *A* and *A*-Er-Eu-U in the 750–3800 cm^−1^ range. Sorption conditions: 333 mg/L mixture of equal amounts of Er^3+^, Eu^3+^, and UO_2_^2+^; 3 mL of 1 wt/v% *A*.

**Figure 4 polymers-15-04457-f004:**
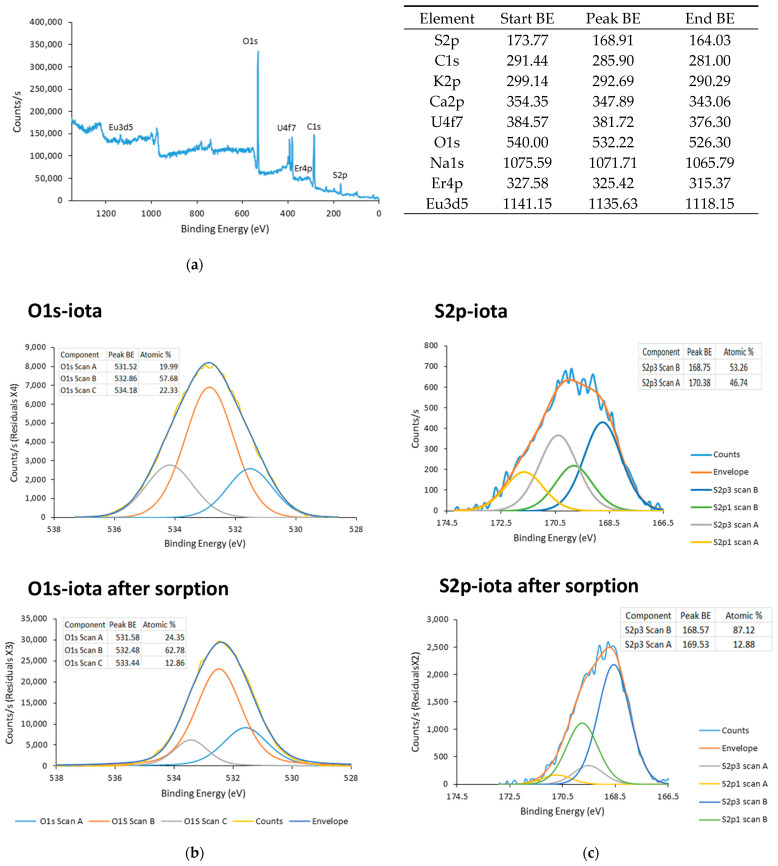
XPS survey spectrum of *I*-Er-Eu-U (**a**). High-resolution O1s peaks of iota before and after sorption and curve fitting (**b**). High-resolution S2p peaks of iota before and after sorption and curve fitting (**c**).

**Table 1 polymers-15-04457-t001:** Sorption yields of different polysaccharides with various metals ^a^.

Entry	Metal Ion (mg/L)/Polysaccharide (g)	Yields (%)	
		*A*	*I*
Eu^3+^	700/0.03	76	68
Sm^3+^	700/0.03	71	66
UO_2_^2+^	700/0.03	99	79
Eu^3+^	500/0.03	96	76
Sm^3+^	500/0.03	97	80
UO_2_^2+^	500/0.03	99	81

^a^ Sorption conditions: 25 °C, 30 min, pH ~5.

**Table 2 polymers-15-04457-t002:** Sorption yields from solutions with Eu^3+^ and Sm^3+ a^.

Eu^3+^:Sm^3+^ (mg/L)	*A (%)*	*I (%)*
350:350	76	66
235:465	70	62
465:235	74	62

^a^ Sorption conditions: 25 °C, 30 min, pH ~5.

**Table 3 polymers-15-04457-t003:** Sorption yields and selectivities (S) of solutions of Eu^3+^ and UO_2_^2+ a^.

Eu^3+^:UO_2_^2+^ (mg/L)	*A*		λ		*I*	
	Y (%)	S (%) Eu^3+^:UO_2_^2+^	Y (%)	S (%) Eu^3+^:UO_2_^2+^	Y (%)	S (%) Eu^3+^:UO_2_^2+^
800:200	83	76:24	65	87:13	56	70:30
500:500	96	50:50	78	59:41	69	60:40
200:800	98	20:80	76	25:75	81	22:78

^a^ Sorption conditions: 0.03 g polysaccharide, 25 °C, 30 min, pH ~5.

**Table 4 polymers-15-04457-t004:** Sorption yields and selectivities of solutions containing Er^3+^ and UO_2_^2+ a^.

Er^3+^:UO_2_^2+^ (mg/L)	*A*		λ		*I*	
	Y (%)	S (%) Er^3+^:UO_2_^2+^	Y (%)	S (%) Er^3+^:UO_2_^2+^	Y (%)	S (%) Er^3+^:UO_2_^2+^
800:200	83	76:24	57	85:15	52	84:16
500:500	95	49:51	75	58:42	56	65:35
200:800	99	19:81	74	24:76	79	24:76

^a^ Sorption conditions: 0.03 g polysaccharide, 25 °C, 30 min, pH ~5.

**Table 5 polymers-15-04457-t005:** Sorption yields and selectivities of solutions with Eu^3+^, Er^3+^, and UO_2_^2+ a^.

	*A*	λ	*I*
Overall Y (%)	85	59	61
UO_2_^2+^ Y (%)	37	21	22
Eu^3+^ Y (%)	31	40	41
Er^3+^ Y (%)	32	39	38

^a^ Sorption conditions: 0.03 g polysaccharide; 333 mg/L Eu^3+^, Er^3+^, and UO_2_^2+^; 25 °C; 30 min; pH ~5.

**Table 6 polymers-15-04457-t006:** Thermal properties of the polysaccharides before and after the addition of Er^3+^, Eu^3+^, and UO_2_^2+^ adsorption mixture.

STEP	Pristine *A*			*A*-Er-Eu-U		
	Onset Temp (°C)	DTG (°C)	Wt Loss (%)	Onset Temp (°C)	DTG (°C)	Wt Loss (%)
Step 1	-	72.70	14.98	-	64.77	12.94
Step 2	150.67	241.02	36.62	149.64	239.35	31.72
Step 3	299.40	435.20	11.24	348.99	400.61	12.32
Step 4	562.10	-	3.69	600.41	855.10	16.33
Residue			33.47			26.69
	**Pristine λ**			**λ-Er-Eu-U**		
Step 1	-	75.51	11.80	-	70.54	14.98
Step 2	181.62	229.07	44.80	137.60	183.07	26.73
Step 3	599.85	-	17.10	299.36	350.67	9.68
Step 4	-	-	-	427.10	488.02, 573.30	13.36
Step 5	-	-	-	650.04	810.00	7.76
Residue			26.30			27.49
	**Pristine *I***			** *I* ** **-Er-Eu-U**		
Step 1	-	87.44	6.92	-	73.82	12.21
Step 2	137.24	164.84	25.02	126.86	148.76	9.57
Step 3	184.98	256.15	12.81	171.61	202.09	19.59
Step 4	307.36	351.05	12.11	307.49	359.39	9.67
Step 5	599.95	747.71	16.60	450.69	514.49, 594.19	12.57
Step 6	-	-	-	650.04	820.01	7.88
Residue			26.54			28.51

## Data Availability

Data are contained within the article.

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
