# Peer review of "Selective Sorption of Heavy Metals by Renewable Polysaccharides"

_polymers, 2023, doi:10.3390/polym15224457_

Round 1
Reviewer 1 Report
Comments and Suggestions for Authors
In general, the article is very interesting and good, but the recommendations would be to supplement the introduction by mentioning the latest findings of other authors specifically about the selected polysaccharides and their metal binding abilities. For now, the emphasis is on the very previous research, which is also interesting.
It would be recommended to substantiate the renewability of polysaccharides more in the article.
The results should be compared more with the results of other authors, for example, how the effectiveness of the polysaccharides used in this study compares with other materials when studying the binding abilities of similar compounds.
Reviewer 2 Report
Comments and Suggestions for Authors
In this manuscript “Selective Sorption of Heavy Metals by Renewable Polysaccharides ”, these three polysaccharides were employed to recover different heavy metal ions, alone or in different mixtures, from aqueous solutions. And the highest sorption yields were obtained for the alginate for all the metal ions, alone or in mixtures. The research is meaningful, but there are some problems with your manuscript. The comments and problems are as follows:
1.In the introduction, it is important to explain what you bring to the table in order to express your originality. The purpose and significance of the study needs to be reflected in the last paragraph. Please add further information and reasons and modify accordingly.
2.The subsection of the manuscript materials and methods section suggests adding serial numbers, such as 2.1,2.2, etc. , to make the manuscript as a whole more rational.
3.There is no space between the number and ℃ in the manuscript.
4.Suggest uniform format of tables in manuscript.
5.Figure 5 (b), (c) in the manuscript need further improvement.
6.The research and results of this paper are meaningful, but what can we get, what can we develop in, and what contribution to this field need to be presented.
Reviewer 3 Report
Comments and Suggestions for Authors
The article “Selective Sorption of Heavy Metals by Renewable Polysaccharides” is interesting and well written. In this article, they studied the adsorption of metal ions by using various polysaccharides and found that alginates have a higher sorption capacity compared to other polysaccharides. The introduction of the article was well written and based on the review of previous literature. The methods were well described in the article. Furthermore, they presented the results in a good format and ensured a relevant discussion. However, I have some suggestions to improve the article, please see below.
1. Line 36: Please provide the name of the metal ion if you mention it for the first time in the article. Also for the other metal ions, which appear for the first time in the article.
2. Introduction: Please briefly discuss what other methods are used to extract metal ions and why you chose these polysaccharides for this study.
3. Line 771: "One g of polysaccharide was dissolved in 100 mL DDW and mixed for 1 h,“: I would suggest writing 1 g instead of “one g”. What is DDW, please expand it.
4. Line 71: “Sorption experiments” – Is there a reference for these experiments?
5. Table 1: It appears that they did not use a control or replicates in their experiments. Please indicate how many replicates you used in these experiments and calculate the SD accordingly.
6. Line 152: “The metal ion sorption tests also showed that the sorption yield of UO22+ on A is higher than that of Sm3+ or Eu3+ (Table 1).“ How big is the difference and is it significant? Please do a significance test.
7. Table 2: How did you choose these ratios “of Eu3+:Sm3+”? And are these the values of A and I or in percent? Please mention it in the table. I also suggest writing the name instead of A and I.
8. Table 3: What is S here in the table? Please mention it in the legend. How many replicates did you use for these experiments and did you use a control?
Comments on the Quality of English LanguageThe English language of the article is fine, some sentences need to be changed.
Reviewer 4 Report
Comments and Suggestions for Authors
The manuscript by Levy-Ontman and co-workers reports a study of sorption of some heavy metals (Eu3+, Er3+, Sm3+ and UO22+) by three negatively charged polysaccharides from renewable sources (alginate and i- and l-carrageenan), in continuation with similar investigations recently reported by the same authors (reff. 11-13). I find the manuscript well written, data are robust, clearly presented and fully supporting the conclusions of the work. Although the degree of novelty is not very high, I judge the manuscript worth being accepted for publication on Polymers, provided the authors amend the few, minor points listed here below:
- among the key structural features of any polysaccharide, there is the molecular weight. Therefore, in the Materials section (L66-L68) the authors should provide this information for each of the three investigated polysaccharides. Moreover, in the case of alginate, the ability to form complexes with metal ions is primarily determined by its relative content of D-mannuronic and L-guluronic acid constituents. This information should be also added;
- data related to sorption experiments are reported in Tables 1-5 as single values although the authors state (L84-85) that each experiment was performed five times. Therefore, standard deviations should be added to each value of Tables 1-5;
- in Materials and Methods section, general information on FT-IR experiments are missing;
- the more stable chair conformation of L-guluronic acid residues in alginates is the 1C4, not the 4C1 as depicted in figure 1;
- L139-141: the authors state that alginate carboxylic groups are more available for complexation than carrageenan sulfate groups due to the simpler structure of the former polysaccharide. What does it mean?
- L196-L198: I cannot catch from data of Table 3 the linear trend for the two carrageenans as indicated by the authors;
- L304-306: I find this sentence off-topic in this paragraph;
- please double-check the correspondence between the values reported within the text at pag.12 and data of Table 6. Did the authors mix some onset temperatures and DTG values up?
Finally, a general consideration concerning the structure-sorption activity relationships of the investigated polysaccharides. Have the authors tested any polysaccharide containing both sulfate and carboxylate groups as anionic moieties? It would be very useful to compare, for example, the sorption activity of carboxylate-containing, natural polysaccharides such as alginate, xanthan gum, pectin etc. with sulfated derivatives thereof obtained by chemical sulfation of their hydroxyl groups.
Comments on the Quality of English LanguageEnglish language is fine. A minor editing is recommended in order to amend few misprints.
Round 2
Reviewer 3 Report
Comments and Suggestions for Authors
I have reviewed the revised version of the paper and some of my suggestions have been implemented. I agree with the author's changes. I have no further suggestions.
Comments on the Quality of English LanguageThe English language of the manuscript is fine, some sentences need to be changed.